# Discrete Flows: Invertible Generative Models of Discrete Data

**Dustin Tran**[1]    **Keyon Vafa**[12][*]    **Kumar Krishna Agrawal**[1][†]    **Laurent Dinh**[1]    **Ben Poole**[1]
[1]Google Brain    [2]Columbia University

## Abstract

While normalizing flows have led to significant advances in modeling high-dimensional continuous distributions, their applicability to discrete distributions remains unknown. In this paper, we show that flows can in fact be extended to discrete events—and under a simple change-of-variables formula not requiring log-determinant-Jacobian computations. Discrete flows have numerous applications. We display proofs of concept under two flow architectures: discrete autoregressive flows that enable bidirectionality, allowing for example tokens in text to depend on both left-to-right and right-to-left contexts in an exact language model; and discrete bipartite flows that enable efficient non-autoregressive generation as in RealNVP.

## 1 Introduction

There have been many recent advances in normalizing flows, a technique for constructing high-dimensional continuous distributions from invertible transformations of simple distributions (Rezende and Mohamed, 2015; Tabak and Turner, 2013; Rippel and Adams, 2013). Applications for high-dimensional continuous distributions are widespread: these include latent variable models with expressive posterior approximations (Rezende and Mohamed, 2015; Ranganath et al., 2016; Kingma et al., 2016), parallel image generation (Dinh et al., 2017; Kingma and Dhariwal, 2018), parallel speech synthesis (Oord et al., 2017; Prenger et al., 2018), and general-purpose density estimation (Papamakarios et al., 2017).

Normalizing flows are based on the change-of-variables formula, which derives a density given an invertible function applied to continuous events. There have not been analogous advances for discrete distributions, where flows are typically thought to not be applicable. Instead, most research for discrete data has focused on building either latent-variable models with approximate inference (Bowman et al., 2015), or increasingly sophisticated autoregressive models that assume a fixed ordering of the data (Bengio et al., 2003; Vaswani et al., 2017). In this paper, we present an alternative for flexible modeling of discrete sequences by extending continuous normalizing flows to the discrete setting. We demonstrate proofs of concept of discrete flows with two architectures:

1. **Discrete autoregressive flows** enable multiple levels of autoregressivity. For example, one can design a bidirectional language model of text where each token depends on both left-to-right and right-to-left contexts while maintaining an exact likelihood and sampling.

2. **Discrete bipartite flows** (i.e., with flow structure similar to RealNVP (Dinh et al., 2017)) enable flexible models with parallel generation. For example, one can design nonautoregressive text models which maintain an exact likelihood for training and evaluation.

### 1.1 Related Work

**Bidirectional models.** Classically, bidirectional language models have been pursued but require approximate inference (Mnih and Teh, 2012). Unlike bidirectional models, autoregressive models must impose a specific ordering, and this has been shown to matter across natural language

---

[*]Work done as an intern at Google Brain. Supported by NSF grant DGE-1644869.
[†]Work done as an AI resident.

processing tasks (Vinyals et al., 2015; Ford et al., 2018; Xia et al., 2017). Bidirectionality such as in encoders have been shown to significantly improve results in neural machine translation (Britz et al., 2017). Most recently, BERT has shown bidirectional representations can significantly improve transfer tasks (Devlin et al., 2018). In this work, discrete autoregressive flows enable bidirectionality while maintaining the benefits of a (tractable) generative model.

**Nonautoregressive models.** There have been several advances for flexible modeling with nonautoregressive dependencies, mostly for continuous distributions (Dinh et al., 2014; 2017; Kingma and Dhariwal, 2018). For discrete distributions, Reed et al. (2017) and Stern et al. (2018) have considered retaining blockwise dependencies while factorizing the graphical model structure in order to simulate hierarchically. Gu et al. (2018) and Kaiser et al. (2018) apply latent variable models for fast translation, where the prior is autoregressive and the decoder is conditionally independent. Lee et al. (2018) adds an iterative refinement stage to initial parallel generations. In this work, discrete bipartite flows enable nonautoregressive generation while maintaining an exact density—analogous to RealNVP advances for image generation (Dinh et al., 2017).

## 2 BACKGROUND

### 2.1 NORMALIZING FLOWS

Normalizing flows transform a probability distribution using an invertible function (Tabak and Turner, 2013; Rezende and Mohamed, 2015; Rippel and Adams, 2013). Let $\mathbf{x}$ be a $D$-dimensional continuous random variable whose density can be computed efficiently. Given an invertible function $f : \mathbb{R}^D \to \mathbb{R}^D$, the change-of-variables formula provides an explicit construction of the induced distribution on the function's output, $\mathbf{y} = f(\mathbf{x})$:

$$p(\mathbf{y}) = p(f^{-1}(\mathbf{y})) \det \left| \frac{d\,\mathbf{x}}{d\,\mathbf{y}} \right|. \tag{1}$$

The transformation $f$ is referred to as a flow and $\mathbf{x}$ is referred to as the base distribution. Composing multiple flows can induce further complex distributions.

### 2.2 FLOW TRANSFORMATION

For an arbitrary invertible $f$, the determinant of the Jacobian incurs an $O(D^3)$ complexity, which is as costly as modeling with a full rank covariance matrix. Thus, normalizing flows are designed so that the determinant of the flow's Jacobian can be computed efficiently. Here, we review two popular flow transformations.

**Autoregressive flows.** Autoregressive functions such as recurrent neural networks and Transformers (Vaswani et al., 2017) have been shown to successfully model data across modalities. Specifically, assume a base distribution $\mathbf{x} \sim p(\mathbf{x})$. With $\boldsymbol{\mu}$ and $\boldsymbol{\sigma}$ as autoregressive functions of $\mathbf{y}$, i.e. $\boldsymbol{\mu}_d, \boldsymbol{\sigma}_d = f(\mathbf{y}_1, \ldots, \mathbf{y}_{d-1})$, and $\boldsymbol{\sigma}_d > 0$ for all $d$, the flow computes a location-scale transform (Papamakarios et al., 2017),

$$\mathbf{y}_d = \boldsymbol{\mu}_d + \boldsymbol{\sigma}_d \cdot \mathbf{x}_d \qquad \text{for } d \text{ in } 1, \ldots, D.$$

The transformation is invertible and in fact, the inverse can be vectorized and computed in parallel:

$$\mathbf{x}_d = \boldsymbol{\sigma}_d^{-1}(\mathbf{y}_d - \boldsymbol{\mu}_d) \qquad \text{for } d \text{ in } 1, \ldots, D$$

In addition to a fast-to-compute inverse, the autoregressive flow's Jacobian is lower-triangular, so its determinant is the product of the diagonal elements, $\prod_{d=1} \sigma_d$. This enables autoregressive flow models to have efficient log-probabilities for training and evaluation.

**Bipartite flows.** Real-valued non-volume preserving (RealNVP) flows are another transformation (Dinh et al., 2017). For some $d < D$, RealNVP coupling flows follow a bipartite rather than

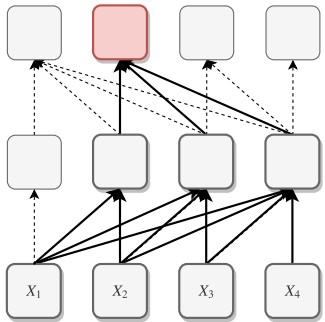 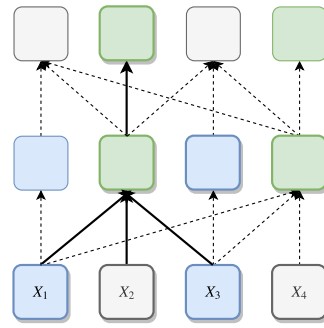

**Figure 1:** Flow transformation when computing log-likelihoods. **(a)** Discrete autoregressive flows stack multiple levels of autoregressivity. The receptive field of output unit 2 (red) includes left and right contexts. **(b)** Discrete bipartite flows apply a binary mask (blue and green) which determines the subset of variables to transform. With 2 flows, the receptive field of output unit 2 is $\mathbf{x}_{1:3}$.

autoregressive factorization:

$$\mathbf{y}_{1:d} = \mathbf{x}_{1:d} \tag{2}$$

$$\mathbf{y}_{d+1:D} = \boldsymbol{\mu} + \boldsymbol{\sigma} \cdot \mathbf{x}_{(d+1):D}, \tag{3}$$

where $\boldsymbol{\sigma}$ and $\boldsymbol{\mu}$ are functions of $\mathbf{x}_{1:d}$ with $\boldsymbol{\sigma} > 0$. By changing the ordering of variables between each flow, the composition of RealNVP flows can learn highly flexible distributions. RealNVP flows have a lower-triangular Jacobian where its determinant is again the product of diagonal elements, $\prod_{i=d+1}^{D} \sigma_i$.

RealNVP flows are not as expressive as autoregressive flows, as a subset of variables don't undergo a transformation. However, both their forward and inverse computations are fast to compute, making them suitable for generative modeling where fast generation is desired.

## 3 DISCRETE FLOWS

Normalizing flows depend on the change of variables formula (Equation 1) to compute the change in probability mass for the transformation. However, the change of variables formula applies only to continuous random variables. We extend normalizing flows to discrete events.

### 3.1 DISCRETE CHANGE OF VARIABLES

Let $\mathbf{x}$ be a discrete random variable and $\mathbf{y} = f(\mathbf{x})$ where $f$ is some function of $\mathbf{x}$. The induced probability mass function of $\mathbf{y}$ is the sum over the pre-image of $f$:

$$p(\mathbf{y} = y) = \sum_{x \in f^{-1}(y)} p(\mathbf{x} = x),$$

where $f^{-1}(y)$ is the set of all elements such that $f(x) = y$. For an invertible function $f$, this simplifies to

$$p(\mathbf{y} = y) = p(\mathbf{x} = f^{-1}(y)). \tag{4}$$

Note Equation 4's relationship to the continuous change of variables formula (Equation 1). It is the same but without the log-determinant-Jacobian. Intuitively, the log-determinant-Jacobian corrects for changes to the volume of a continuous space; volume does not exist for discrete distributions so there is no need to adjust it. Computationally, Equation 4 is appealing as there are no restrictions on $f$ such as fast Jacobian computations in the continuous case, or tradeoffs in how the log-determinant-Jacobian influences the output density compared to the base distribution.

### 3.2 DISCRETE FLOW TRANSFORMATION

Next we develop discrete invertible functions. To build intuition, first consider the binary case. Given a $D$-dimensional binary vector $\mathbf{x}$, one natural function applies the XOR bitwise opera-

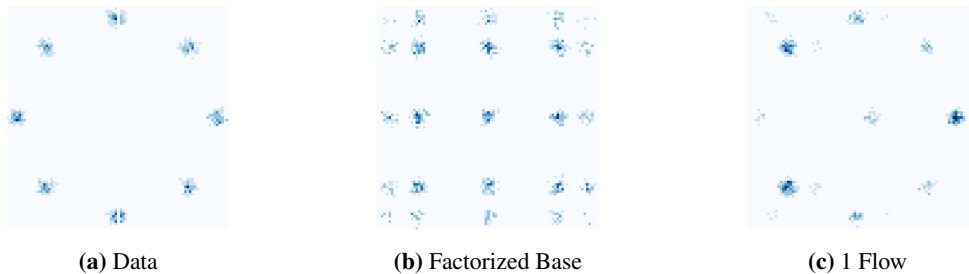

**(a)** Data          **(b)** Factorized Base          **(c)** 1 Flow

**Figure 2:** Learning a discretized mixture of Gaussians with maximum likelihood. Discrete flows help capture the multi-dimensional modes, which a factorized distribution cannot. (Note because the data is 2-D, discrete autoregressive flows and discrete bipartite flows are equivalent.)

tor,

$$\mathbf{y}_d = \boldsymbol{\mu}_d \oplus \mathbf{x}_d, \qquad\qquad \text{for } d \text{ in } 1, \dots, D,$$

where $\boldsymbol{\mu}_d$ is a function of previous outputs, $\mathbf{y}_1, \dots, \mathbf{y}_{d-1}$; $\oplus$ is the XOR function (0 if $\boldsymbol{\mu}_d$ and $\mathbf{x}_d$ are equal and 1 otherwise). The inverse is $\mathbf{x}_d = \boldsymbol{\mu}_d \oplus \mathbf{y}_d$. We provide an example next.

**Example.** Let $D = 2$ where $p(\mathbf{x})$ is defined by the following probability table:

|              | $\mathbf{x}_2 = 0$ | $\mathbf{x}_2 = 1$ |
|--------------|--------------------|--------------------|
| $\mathbf{x}_1 = 0$ | 0.63 | 0.07 |
| $\mathbf{x}_1 = 1$ | 0.03 | 0.27 |

The data distribution cannot be captured by a factorized one $p(\mathbf{x}_1)p(\mathbf{x}_2)$. However, it can with a flow: set $f(\mathbf{x}_1, \mathbf{x}_2) = (\mathbf{x}_1, \mathbf{x}_1 \oplus \mathbf{x}_2)$; $p(\mathbf{x}_1)$ with probabilities $[0.7, 0.3]$; and $p(\mathbf{x}_2)$ with probabilities $[0.9, 0.1]$. The flow captures correlations that cannot be captured alone with the base. More broadly, discrete flows perform a multi-dimensional relabeling of the data such that it's easier to model with the base. This is analogous to continuous flows, which whiten the data such that it's easier to model with the base (typically, a spherical Gaussian).

**Modulo location-scale transform.** To extend XOR to the categorical setting, consider a $D$-dimensional vector $\mathbf{x}$, each element of which takes on values in $0, 1, \dots, K - 1$. One can perform location-scale transformations on the *modulo integer space*,

$$\mathbf{y}_d = (\boldsymbol{\mu}_d + \boldsymbol{\sigma}_d \cdot \mathbf{x}_d) \bmod K. \tag{5}$$

Here, $\boldsymbol{\mu}_d$ and $\boldsymbol{\sigma}_d$ are autoregressive functions of $\mathbf{y}$ taking on values in $0, 1, \dots, K-1$ and $1, \dots, K-1$ respectively. For this transformation to be invertible, $\boldsymbol{\sigma}$ and $K$ must be coprime (an explicit solution for $\sigma^{-1}$ is Euclid's algorithm). An easy way to ensure coprimality is to set $K$ to be prime; mask noninvertible $\boldsymbol{\sigma}$ values for a given $K$; or fix $\boldsymbol{\sigma} = 1$. Setting $K = 2$ and $\boldsymbol{\sigma} = 1$, it's easy to see that the modulo location-scale transform generalizes XOR. The idea also extends to the bipartite flow setting: the functions $(\boldsymbol{\mu}, \boldsymbol{\sigma})$ are set to $(0, 1)$ for a subset of the data dimensions, and are functions of that subset otherwise.

**Example.** Figure 2 illustrates an example of using flows to model correlated categorical data. Following Metz et al. (2016), the data is drawn from a mixture of Gaussians with 8 means evenly spaced around a circle of radius 2. The output variance is $0.01$, with samples truncated to be between $-2.25$ and $2.25$, and we discretize at the $0.05$ level. A factorized base distribution cannot capture the data correlations, while a single discrete flow can. (Note the modulo location-scale transform does not make an ordinal assumption. We display ordinal data as an example only for visualization; other experiments use non-ordinal data.)

### 3.3 TRAINING DISCRETE FLOWS

With discrete flow models, the maximum likelihood objective per datapoint is

$$\log p(\mathbf{y}) = \log p(f^{-1}(\mathbf{y})),$$

|  | Autoregressive Base | Autoregressive Flow | Factorized Base | Bipartite Flow |
|---|---|---|---|---|
| $D = 2, K = 2$ | **0.9** | **0.9** | 1.3 | **1.0** |
| $D = 5, K = 5$ | 7.7 | **7.6** | 8.0 | **7.9** |
| $D = 5, K = 10$ | 10.7 | **10.3** | 11.5 | **10.7** |
| $D = 10, K = 5$ | 15.9 | **15.7** | 16.6 | **16.0** |

**Table 1:** Negative log-likelihoods for the full rank discrete distribution (lower is better). Autoregressive flows improve over its autoregressive base. Bipartite flows improve over its factorized base and achieve nats close to an autoregressive distribution while remaining parallel.

|  | Autoregressive Base | Autoregressive Flow |
|---|---|---|
| $D = 16$, coupling $= 0.5$ | **0.73** | **0.73** |
| $D = 25$, coupling $= 0.1$ | 1.62 | **1.12** |
| $D = 25$, coupling $= 0.5$ | 0.92 | **0.92** |
| $D = 25$, coupling $= 1.0$ | 7.54 | **4.53** |
| $D = 25$, coupling $= 1.5$ | 15.05 | **5.81** |

**Table 2:** Negative log-likelihoods on the square-lattice Ising model (lower is better). Higher coupling strength corresponds to more spatial correlations.

where the flow $f$ has free parameters according to its autoregressive or bipartite network, and the base distribution $p$ has free parameters as a factorized (or itself an autoregressive) distribution. Gradient descent with respect to base distribution parameters is straightforward. To perform gradient descent with respect to flow parameters, one must backpropagate through the discrete-output function $\boldsymbol{\mu}$ and $\boldsymbol{\sigma}$. We use the straight-through gradient estimator (Bengio et al., 2013). In particular, the (autoregressive or bipartite) network outputs two vectors of $K$ logits $\theta_d$ for each dimension $d$, one for the location and scale respectively. On the forward pass, we take the argmax of the logits, where for the location,

$$\boldsymbol{\mu}_d = \text{one\_hot}(\text{argmax}(\theta_d)). \tag{6}$$

Because the argmax operation is not differentiable, we replace Equation 6 on the backward pass with the softmax-temperature function:

$$\frac{d\boldsymbol{\mu}_d}{d\theta_d} \approx \frac{d}{d\theta_d} \text{softmax}\left(\frac{\theta_d}{\tau}\right).$$

As the temperature $\tau \to 0$, the softmax-temperature becomes close to the argmax and the bias of the gradient estimator disappears. However, when $\tau$ is too low, the gradients vanish, inhibiting the optimization. Work with the Gumbel-softmax distribution indicates that this approximation works well when the number of classes $K < 200$ (Maddison et al., 2016; Jang et al., 2017), which aligns with our experimental settings; we also fix $\tau = 0.1$.

## 4 TOY EXPERIMENTS

In addition to the experiment in Figure 2, we perform three toy experiments to show the utility of discrete autoregressive flows and discrete bipartite flows. For discrete autoregressive flows, we used an autoregressive Categorical base distribution where the first flow is applied in reverse ordering. (This setup lets us compare its advantage of bidirectionality to the baseline of an autoregressive base with 0 flows.) For discrete bipartite flows, we used a factorized Categorical base distribution where the bipartite flows alternate masking of even and odd dimensions.

**Full-rank Discrete Distribution.** A natural experiment is to analyze the expressivity of the flows for an arbitrary discrete distribution. In particular, we sample a true set of probabilities for all $D$ dimensions of $K$ classes according to a Dirichlet distribution of size $K^D - 1$, $\alpha = 1$. For the network for both the base and flows, we used a Transformer with 64 hidden units.

Table 1 displays negative log-likelihoods (nats) of trained models over data simulated from this distribution. Across the data dimension $D$ and number of classes $K$, autoregressive flows gain several nats over the autoregressive base distribution, which has no flow on top. Bipartite flows improve over its factorized base and in fact obtain nats competitive with the autoregressive base while remaining fully parallel for generation.

**Addition.** Following Zaremba and Sutskever (2014), we examine an addition task: there are two input numbers with $D$ digits (each digit takes $K = 10$ values), and the output is their sum with $D$ digits (we remove the $D + 1^{th}$ digit if it appears). Addition naturally follows a right-to-left ordering: computing the leftmost digit requires carrying the remainder from the rightmost computations. Given an autoregressive base which poses a left-to-right ordering, we examine whether the bidirectionality that flows offer can adjust for wrong orderings. We use an LSTM to encode both inputs, apply 0 or 1 flows on the output, and then apply an LSTM to parameterize the autoregressive base where its initial state is set to the concatenated two encodings. All LSTMs use 256 hidden units for $D = 10$; 512 for $D = 20$.

For $D = 10$, an autoregressive base achieves **4.0** nats; an autoregressive flow achieves **0.2** nats (i.e., close to the true deterministic solution over all pair of 10-digit numbers). A bipartite model with 1, 2, and 4 flows achieves **4.0**, **3.17**, and **2.58** nats respectively. For $D = 20$, an autoregressive base achieves **12.2** nats; an autoregressive flow achieves **4.8** nats. A bipartite model with 1, 2, 4, and 8 flows achieves **12.2**, **8.8**, **7.6**, and **5.08** nats respectively.

**Ising Model.** We examine how bidirectional generative models can be used for learning undirected models. For the base network, we used a single layer LSTM with 8 hidden units. For the flow network, we used an embedding layer with 8 hidden units.

Table 2 displays negative log-likelihoods (nats) of trained models over data simulated from Ising models with varying lattice size and coupling strength. As Ising models are undirected models, the autoregressive base posits a poor inductive bias by fixing an ordering and sharing network parameters across the individual conditional distributions. Over data dimension $D$ and coupling, autoregressive flows perform as well as, or improve upon, autoregressive base models.

## 5  LIMITATIONS

We describe discrete flows, a class of invertible functions for flexible modeling of discrete data. Note our experiments are only toy to show proofs of concept. We're continuing to push these ideas to larger-scale text data. We're also applying discrete inverse autoregressive flows, which enable flexible variational approximations for discrete latent variable models. One open question remains with scaling discrete flows to large numbers of classes: in particular, the straight-through gradient estimator works well for small numbers of classes such as for character-level language modeling, but it may not work for (sub)word-level modeling where the vocabulary size is greater than 5,000.

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
