# OpenReview forum: "Discrete Flows: Invertible Generative Models of Discrete Data"
_ICLR.cc/2019/Workshop/DeepGenStruct — DeepGenStruct 2019_

### Official Review · AnonReviewer2 · 2019-04-15
**Normalizing flows for discrete distributions**

**Rating:** 3
**Confidence:** 1

**Review:**

The paper extends the concept of normalizing flows in modeling high-dimensional continuous distributions to discrete distributions.
It is true that the paper only presents a proof of concept with few toy experiments,  it is well written and does a good job motivating discrete flows.  However, a real application of these models to model large-scale text data with large vocabulary is still an open question.

---

### Official Review · AnonReviewer1 · 2019-04-16
**Interesting preliminary work**

**Rating:** 3
**Confidence:** 2

**Review:**

This paper proposes a version of normalizing flows applicable to discrete data. The authors motivate this idea in two ways: (1) autoregressive models that are bidirectional, and (2) non-autoregressive likelihood-based models of discrete data. They show that flows on discrete data do not need an expensive log-det-Jacobian step. However, they require a function mapping from discrete variables to discrete variables. As such a function is not directly differentiable, this paper uses a straight-through gradient estimator.

This is an interesting idea but is not that well fleshed out. A more thorough discussion of calculating gradients through the discrete outputs of their neural networks seems in order. Additionally, the experiments shown are on very small toy data.

Pros:
- Interesting idea
- Clear, no-nonsense writing

Cons:
- No experiments on real data
- Straight-through estimator seems limiting (e.g. not sure how it applies to non-ordinal data)

---

### Decision · Program_Chairs · 2019-04-19
**Acceptance Decision**

Accept